# Characterization of Calpain and Caspase-6-Generated Glial Fibrillary Acidic Protein Breakdown Products Following Traumatic Brain Injury and Astroglial Cell Injury

**DOI:** 10.3390/ijms23168960

**Published:** 2022-08-11

**Authors:** Zhihui Yang, Rawad Daniel Arja, Tian Zhu, George Anis Sarkis, Robert Logan Patterson, Pammela Romo, Disa S. Rathore, Ahmed Moghieb, Susan Abbatiello, Claudia S. Robertson, William E. Haskins, Firas Kobeissy, Kevin K. W. Wang

**Affiliations:** 1Program for Neurotrauma, Neuroproteomics and Biomarkers Research, Department of Psychiatry, McKnight Brain Institute, University of Florida, Gainesville, FL 32611, USA; 2The Departments of Psychiatry, University of Florida, Gainesville, FL 32611, USA; 3Department of Pediatrics, Daping Hospital, Third Military Medical University, Chongqing 400038, China; 4The Departments of Chemistry, University of Florida, Gainesville, FL 32611, USA; 5National Laboratory, Biological Sciences Division/Integrative Omics, Pacific Northwest, 902 Battelle Boulevard, Richland, WA 99352, USA; 6The Barnett Institute of Chemical and Biological Analysis, Northeastern University, 360 Huntington Ave, Boston, MA 02115, USA; 7Department of Neurosurgery, Baylor College of Medicine, Houston, TX 77030, USA; 8Gryphon Bio, Inc., 611 Gateway Blvd. Suite 120 #253, South San Francisco, CA 94080, USA

**Keywords:** astroglial injury, GFAP, calpain, caspase, biomarkers, traumatic brain injury

## Abstract

Glial fibrillary acidic protein (GFAP) is the major intermediate filament III protein of astroglia cells which is upregulated in traumatic brain injury (TBI). Here we reported that GFAP is truncated at both the C- and N-terminals by cytosolic protease calpain to GFAP breakdown products (GBDP) of 46-40K then 38K following pro-necrotic (A23187) and pro-apoptotic (staurosporine) challenges to primary cultured astroglia or neuron-glia mixed cells. In addition, with another pro-apoptotic challenge (EDTA) where caspases are activated but not calpain, GFAP was fragmented internally, generating a C-terminal GBDP of 20 kDa. Following controlled cortical impact in mice, GBDP of 46-40K and 38K were formed from day 3 to 28 post-injury. Purified GFAP protein treated with calpain-1 and -2 generates (i) major N-terminal cleavage sites at A-56*A-61 and (ii) major C-terminal cleavage sites at T-383*Q-388, producing a limit fragment of 38K. Caspase-6 treated GFAP was cleaved at D-78/R-79 and D-225/A-226, where GFAP was relatively resistant to caspase-3. We also derived a GBDP-38K N-terminal-specific antibody which only labels injured astroglia cell body in both cultured astroglia and mouse cortex and hippocampus after TBI. As a clinical translation, we observed that CSF samples collected from severe human TBI have elevated levels of GBDP-38K as well as two C-terminally released GFAP peptides (DGEVIKES and DGEVIKE). Thus, in addition to intact GFAP, both the GBDP-38K as well as unique GFAP released C-terminal proteolytic peptides species might have the potential in tracking brain injury progression.

## 1. Introduction

Astrocytes, also called astroglia are a type of glial cells in the brain and spinal cord that also include perivascular microglia, oligodendrocytes, radial glia, and Muller cells. Astroglia are the most abundant cell types in the brain, providing both structural and functional support for neurons, such as neurotransmitter glutamate recycling and trophic factor release. Astroglial cells contain a unique structural protein, glial fibrillary acidic protein (GFAP) [1]. GFAP is an intermediate filament (IF) III protein responsible for maintaining the cytoskeleton structure and mechanical strength of astroglia cells and supporting the blood-brain barrier (BBB) [2]. In addition, in brain injury models, GFAP seems to provide structural integrity to the vasculature through their enveloping end feet [3] and then stabilize the BBB. GFAP is also highly inducible via the STAT3 transcription factor pathway [4] when astroglia cultures are activated by external stimuli such as LPS or glial-derived neurotrophic factor (GDNF) or ciliary neurotrophic factor (CTNF) or following brain or spinal cord injury in vivo (termed astrogliosis) [1]. Activated astroglia cells show thickened and elongated processes that are maintained by GFAP, among other cytoskeletal protein networks (including vimentin) [5].

There are ten isoforms/splice variants identified for GFAP [1,6,7]. In the adult mammalian brain, the most abundant isoform is GFAP-α (alpha). Structurally, GFAP is organized with a head domain (residues 1–72 in human GFAP-α, protein accession #NP_002046.1), followed by the central rod domain (residues 73-377) that is composed of four coils (1A, 1B, 2A, and 2B) which are flanked by three linker regions (1, 1.2, and 2, respectively) and a C-terminal domain (residues 378–432). This structural organization is generally conserved among class III IF proteins [8,9]. Both the N-terminal head domain (including the MERRRITSARRSY motif), and C-terminal tail domain are required for facilitating oligomerization and filament elongation [10]. Moreover, both the head and tail domains contain multiple sites for phosphorylation and citrullination modifications [1].

Neurotrauma such as TBI and spinal cord injury (SCI) is often associated with neuronal injury or death; several studies from our lab and others have investigated and identified several astroglial and neural biomarkers utilizing experimental TBI/SCI models via omics approaches [11,12,13,14]. However, there is evidence that astroglia could also be subjected to mechanical or chemical injury after neurotrauma and also in neurodegenerative disorders such as Alzheimer’s disease [15]. Thus, we propose that astroglial injury also plays a significant role in neurotrauma. Since GFAP is a major inducible protein in astroglia, GFAP and its modified forms might be released into biofluid (e.g., cerebrospinal fluid (CSF) and blood) following neurotrauma. Recently, GFAP elevations in CSF or serum samples have been identified in several rat or mouse models of severe TBI (control cortical impact, penetrating brain injury, blast overpressure wave brain injury) as well as in serum/plasma samples in mild TBI models (close head concussive brain injuries) [16,17,18,19]. In addition, the post-TBI elevation of GFAP appears severity-dependent [20]. In humans, GFAP levels are elevated acutely (within 24–48 h) in CSF and serum/plasma following severe TBI [21,22,23,24] and in serum/plasma samples after moderate to mild TBI, as measured by the sandwich ELISA method [25]. Lastly, GFAP protein levels in CSF serum and/or plasma are sensitive to the severity of TBI as defined by the Glasgow coma scale (GCS), cranial computed tomography (CT) abnormality, magnetic resonance imaging (MRI)-detected pathological alterations, and patient outcomes [25,26,27,28].

GFAP protein has been reported to be potentially sensitive to proteolysis [29,30]. Two responsible protease candidates proposed are calpains and caspases [20,29,31,32,33]. Some recent studies proposed that GFAP breakdown products (GBDP) are released into biofluid after brain injury [25]. Since we and others have previously demonstrated that both calpains and caspases are cell death proteases over-activated in neuronal injury, leading to neuronal protein proteolytic such as αII-spectrin, Tau, collapsin response mediator proteins (CRMPs) [31,32,33,34], we hypothesize that this concept might also carry over to astroglial injury. Thus, in this study, we systemically examined the potential vulnerability of GFAP protein to proteolytic attack and their respective fragmentation patterns under in vitro digestion to calpains and caspases; in cell culture conditions where astroglia cells are injured, in a rodent model of TBI and human CSF samples from TBI patients.

## 2. Results

### 2.1. GFAP Fragmentation in Cultured Neuron-Astroglia Cells

We first examined GFAP protein sensitivity to proteolysis in rat cerebrocortical neuron-astroglial mixed culture. We first compared three different cytotoxic challenges (pro-necrosis calcium ionophore A23187 (10 µM), pro-apoptotic staurosporine (STS, 1 µM), and the excitotoxin glutamate analogue N-methyl-D-aspartate (NMDA, 300 µM). Using a monoclonal anti-GFAP antibody directed to the GFAP protein core, we observed that intact GFAP (50 kDa) was degraded to a 38 kDa major breakdown product (GBDP-38K) [35], as well as several minor intermediate fragments between 48K and 40K in both A23187 and STS treatment, but not NMDA challenge, since astroglia cells are not vulnerable to excitotoxicity (Figure 1A). In our rat mixed culture, since there is a low level of baseline cell death, thus we found that some minor bands of GBDP between 48K and 44K are present even in the control cells. These bands became more intense after A23187 or STS challenges. But these intermediate BDPs (breakdown products) have lower intensity than the limit fragment GBDP-38K following both challenges. In parallel, we also tracked the axonally enriched protein αII-spectrin, which is known to be susceptible to proteolysis. Here, as expected, it was degraded to calpain-generated breakdown product SBDP150/145 (αII-Spectrin Breakdown Products) pair in both A23187 and STS treatment, while caspase-generated SBDP120 was also observed in STS treatment only. In addition, unlike GFAP, αII-spectrin was also degraded to SBDP150/145 pair with NMDA treatment. This is consistent with neuronal cells being sensitive to excitotoxicity, while astroglia cells are not.

Since 6 h of A23187 and STS treatments only resulted in partial cleavages of GFAP as detected with the GFAP core antibody (Figure 1A), we then extended the challenge period to 24 h for further analysis (Figure 1B). There we observed an almost complete conversion of 50K intact GFAP to the 38K GBDP in the A23187 treatment. Similarly, more extensive proteolysis of intact GFAP to GBDP-38K was also observed within 24 h of STS treatment.

Since calpains and caspases are potentially activated in cytotoxic conditions, we also examined the GFAP proteolysis sensitivity to calpain-1, 2 inhibitor MDL28170 or pan-caspase inhibitor Z-D-DCB. When blots were probed with anti-GFAP-core antibodies, we indeed found that A23187 and STS-induced GBDP-38K were inhibited by MDL28170 but not by Z-D-DCB (Figure 1B). We further probed the same cell lysate with a GFAP-C-terminal-directed antibody. In this case, A23187 treatment led to the disappearance of the 50K GFAP, but with only a minor 44K BDP, and no small fragments were observed. Again, MDL28170 restored the level of intact GFAP-50K. In contrast, we observed a very different profile with STS treatment, which produced minor GBDP-44-46K and a major 20K GBDP as detected by the C-terminal antibody. These fragments were inhibited by Z-D-DCB but not MDL28170. Lastly, another pro-apoptotic condition (5 mM EDTA) was added. Due to its sequestration of calcium from intracellular calcium store, EDTA treatment triggers apoptosis, this cytotoxin was known to evoke only caspase activation but not calpain activation [36]. As anticipated, no calpain-generated GBDP-38K was observed with GFAP core antibody, but Z-D-DCB sensitive GBDP-46-44K and GBDP20K were readily observed with GFAP C-terminal antibody (Figure 1B). Lastly, in parallel, as confirmation, we also examined SBDP’s sensitivity to the same protease inhibitors. As expected, SBDP150 /SBDP145 pair (in A23187 and STS treatments) was sensitive to MDL28170, while SBDP120 (in STS and EDTA treatments) was sensitive to Z-D-DCB (Figure 1B). Since caspase-6 has been implicated in GFAP fragmentation [37], we also employed a caspase-6 specific inhibitor Z-IVED-FMK and a pan-caspase inhibitor Z-D-DCB on STS treatment and found that neither of them blocks GBDP-38K formation, while calpain inhibitor MDL28170 did (Appendix A). This further confirms that the dominant GBDP-38K is likely solely derived from calpain-mediated proteolysis.

### 2.2. Generation of the Release of GBDP into Culture Media in Primary Astroglia Cells

Since SBDPs are known not only to be generated in injured neurons but also released into cultured media [38,39], we examined if GBDPs have similar properties upon cell injury. As primary astroglia cells were subjected to A23187 for various time points (0.5, 1,3, 6, 18, and 24 h), cell lysate and cell-conditioned media were collected. In cell lysate, we observed a rapid degradation of GFAP to intermediate fragments of 44K and 46K, followed by the appearance of GBDP-38K (using GFAP core MAb). In cell-conditioned media, the patterns were similar, except that intact GFAP-50K was not observed prominently while GBDP-38K was very prominent, suggesting that GBDPs are selectively released (Figure 2). In parallel, in A23187-treated cells, SBDP150/145 was also observed in cell lysate and cell media at all time points. Again, as expected, intact αII-spectrin was much less prominent in cell media (Figure 2A). When STS was used as a challenge, we visualize both calpain-generated GBDP-38K and caspase-generated GBDP-20K, and we used GFAP core MAb (see Figure 1). A time-dependent formation of minor GBDP-44K-46K (from 1 h to 24 h) and GBDP-38K (from 6 h to 24 h) took place, while intact GFAP was reduced in the cell lysate. In STS-treated cell-conditioned media, again GBDP-44K-46K (from 1–28 h) and GBDP-38K were reduced; intact GFAP was not released in control cells (Figure 2B). We also observed a minor GBDP-22K in the culture media at 18–24 h post-injury, SBDP150/145 and SBDP120 were progressively produced in STS-treated cells and released into cell media. Again, the intact αII-spectrin was not prominently released into cell media (Figure 2B). Third, with EDTA-treated cells, we observed a delayed formation of GBDP-44K (detected with GFAP C-terminal PAb) peaking at 18–24 h. the GBDP-44K was also released into cell media within the same timeframe (Figure 2C). Similarly, the SBDP120 generation and formation parallel those of GBDP-44K.

We further quantified the formation and release of GBDP-38K with A23187- and STS-treated cells and GBDP-44K with EDTA-treated cells into culture media over time (Figure 2E,F), as well as cytosolic LDH release as cell death measurement (Figure 2D). We observed that LDH-release tracked very closely with GBDP formation and release. Cell death as measured by LDH release occurs as early as 30 min to 1 h with A23187 treatment, which is generally reflected by GBDP-38K formation in cells and release into media. LDH release, GBDP-38K formation, and release in STS-treated cells are all more delayed and only become prominent at 18–24 h. Similarly, EDTA caused some early LDH release, but its peak levels of LDH were at 18–24 h. At the same time frame, GBDP-44K was also prominently generated in cells and released into cell media. In addition, we are interested in finding out if GFAP is also proteolyzed under a relatively mild cellular challenge. In this case, we used 200 ng/mL LPS as a challenge, in comparison to A23187 and STS challenges on rat primary glial culture. We were able to observe the calpain-inhibitor-sensitive GBDP-44-46K and GBDP-38K in LPS-treated cells at 24 h, although their levels are much less than those produced with A23187 and STS treatment (Appendix A).

### 2.3. In Vitro GFAP Fragmentation Characterization

To further study GFAP fragmentation, we subjected rat cerebrocortical mixed culture (containing solubilized GFAP) to in vitro digestion with calpain-1, 2, and caspase-3 (Appendix A), as anticipated, GBDP-38K was prominently observed with both calpain treatment (GFAP-core antibody panel) (Appendix A), in contrast, only a very minor GBDP-20K was observed (GFAP C-terminal antibody panel). SBDP profile confirmed that both calpain and caspase-3 are active, generating SBDP150/145 and SBDP150i and SBDP120, respectively [34]. To confirm the relative resistance of GFAP to caspase-3, we next digested purified recombinant full-length GFAP to calpain-1 versus caspase-3 digestion. Again GBDP-38K in calpain treatment was readily observed, but GBDP-20K was not readily detectable in caspase-3 treatment (Appendix A). Chen et al. previously reported that GFAP might be sensitive to caspase-6 [40]. We thus subjected recombinant GFAP to calpain-1 and caspase-6 digestion. With GFAP core antibody, GBDP-44K-46K and GBDP-38K were generated with purified full-length GFAP digested with calpain-1 (Figure 3A). With GFAP C-terminal antibody, calpain digestion generated GBDPs of 46-42K, due to truncations of the C-terminal (Figure 3B). In contrast, caspase-6 was effective in generating an N-terminal GBDP-22K and a C-terminal GBDP-20K from full-length GFAP, respectively (Figure 3A,B).

To identify the cleavage sites of GFAP by calpain and caspase-6, we further subjected digested GFAP-generated fragments to N-terminal sequencing. The results are summarized in Figure 3D (table). With calpain digestion, we observed two major new N-terminals starting at A56GALNAGFKETRASE and A60GFKETRASE based on N-terminal sequencing data of the 38K fragment generated; thus, these place the cleavage between L55*A56 and N59*A60. We also identified two minor N-terminal cleavages (M74*M75ELND and M75*E76LND) (Figure 3D). For C-terminal calpain cleavage, we conducted carboxyl-peptidase digestion and identified the following C-terminal cleavages 1) ITIPVQT383*F384SNLQIRETS; [2] ITIPVQTFSNL387*Q388 IRETS (Figure 3D). With caspase-6 digestion, we identified a 44K fragment that has an R79FASYIEKVRF while a 20K fragment has a new N-terminal of V226AKPDLTAA. Thus, they allow us to map the two major cleavage sites to ELND78*R79FASYIEKVRF and VELD225*V226AKPDLTAA, respectively (Figure 3D). We note that both cleavages were preceded by X-X-X-D motif, consistent with caspase-6 substrate specificity. Taken together, caspase-6 digestion will ultimately generate a 44K intermediate GBDP, an N-terminal 22K GBDP, and the corresponding C-terminal 20K GBDP. We observe all of these with immunoblotting data (Figure 1, Figure 2 and Figure 3A–C). A schematic of GFAP cleavages by calpain-1, 2, and caspase-6 is illustrated in Figure 3E. The GFAP core MAb epitope and GFAP C-terminal PAb epitope are also indicated.

### 2.4. GBDP-38K Specific Antibody and Its Utilities

With the knowledge of the N-terminal cleavage, we raised fragment-specific polyclonal antibodies against one of the new N-terminal of GBDP-38K (A61GFKETRASE). When this GBDP-specific antibody was used to probe against intact purified GFAP, no band was detected, but following calpain digestion, a 38K GBDP was readily detected (Figure 3C). In contrast, caspase-6-generated BDP-20K or GBDP-22K were not detected by this GBDP-specific antibody (Figure 3C).

To further characterize the N-terminal GBDP antibody, in vitro digestion of GFAP in rat cerebrocortical neuron-astroglia mixture lysate with calpain-1 vs. caspase-6 was done. Calpain digestion generated the GBDP-38K as core fragment (panel (ii), Figure 4B) and a N-terminal-truncated transient fragment GBDP-42K (panel (iii), Figure 4B). We note that only the GBDP-38K was uniquely detected by the GBDP antibody (panel (iv), Figure 4B). On the other hand, caspase-6 generated N-terminal GBDP-22K and C-terminal GBDP-20K (Figure 4A). But neither fragment was detected by the GBDP-specific antibody, illustrating its specificity toward calpain-mediated cleavage. Again, calpain produced SBDP150/145 and caspase-6 produced SBDP150I, respectively, as expected (Figure 4A,B) [34].

Importantly, we compared the use of total GFAP core antibody and GBDP-specific antibody in immunocytochemistry studies (Figure 4C). Rat primary astroglia cells were either untreated or treated with A23187, STS, EDTA, or NMDA for 16 h followed by fixing and processing for immunocytochemical staining. GFAP core antibody (MAb 556330) labeled both cell body and healthy processes in control and NMDA conditions but also labeled degenerative astroglia processes and cell body under both pro-necrotic (A23187) and pro-apoptotic (STS, EDTA) cell injury conditions. NMDA did not affect healthy astroglia, thus the cell appearance is similar to untreated cells. In contrast, and importantly, GBDP-specific PAb does not label control astrocytes or EDTA-treated caspase-dominant apoptotic astrocytes, but only labeled injured astroglia cell body and cell processes in cell injury conditions (A23187, STS) where calpain is known to be activated (Figure 4C).

We also confirmed the utilities of the GBDP with immunoblotting of astroglia cell culture treated with A23187 (Appendix A). With A23187, a GBDP-38K band and a minor GBDP-40K band were detected in cell lysate from A23187 treated cells with the GBDP-specific antibody. A strong calpain inhibitor (SNJ-1945) inhibited the appearance of both GBDP-40K and GBDP-38K bands, while caspase inhibitor Z-D did not (Appendix A).

### 2.5. Cytotoxic Effects of GBDP on Rat Cerebrocortical Culture

Since GBDPs are released into culture media and many CNS-generated protein fragments can be cytotoxic (e.g., Aβ peptides, TDP-43 fragment, and Tau fragments), we thus examined if GBDP generated by calpain and by caspase-6 are potentially cytotoxic. Thus, rat cerebrocortical culture lysate was incubated with calpain digested GFAP and caspase-6-digested GFAP or intact GFAP at 100 ng/mL. We found that GFAP protein in cell culture can also evoke cytotoxicity when compared to control culture (* *p* < 0.02) (Figure 5). However, both calpain and caspase-6 digested GFAP appeared more toxic to the culture cells (** *p* < 0.001) (Figure 5).

### 2.6. In Vivo GBDP Generation in a Mouse Model of TBI with Immunoblotting and Immunohistochemical Characterization

We next sought to examine if GFAP fragmentation extends to in vivo traumatic brain injury conditions. We used the established mouse-controlled cortical impact model (CCI) where the focal injury occurs in the cortex unilaterally [41]. Ipsilateral cortex lysate samples from naïve sham or injury (CCI) on days 1, 3, 7, 14, and 28 post-injury were subjected to immunoblotting and probed with GFAP-core MAb. In Figure 6A,B, unlike GFAP itself which already has a strong signal in naïve and sham cortex, GBDPs were barely detectable in naïve or sham cortex. Second, unlike full-length GFAP that rises within 3–7 days, we observed minor levels of GBDPs of 46K to 38K on day 1 post-CCI, but the elevations of GBDP continue over time and did not peak until 14–28 days. Thus, the formation of GBDP is a delayed process that likely reflects a delayed astroglia cell injury or death up to 2–4 weeks following the initial injury event. In parallel, we also directly compared and contrasted the GFAP fragmentation patterns with GFAP core MAb and GBDP-38K N-terminal-specific MAb in hippocampus samples following CCI as compared to naïve and sham samples. Here GBDP-specific MAb did not detect naïve or sham hippocampus but sensitively detected the formation of GBDP-40K and GBDP-38K at D3 and D7 post- CCI samples (Figure 6C). In contrast, the total GFAP antibody detected both full-length GFAP in all naïve, sham and CCI samples as well as its GBDP 44K-38K in the CCI samples (D1, D3, D7). We also noted that the formation of GBDP, in general, paralleled the SBDP formation that mainly occurs in neurons. Thus, GBDP-specific MAb has the potential of uniquely detecting injured astrocytes following brain injury.

We then further characterized the injured cortex by an immunohistochemical (IHC) approach (Figure 7). Ipsilateral and contralateral cortex samples were examined by staining with total GFAP (GFAP core MAb) as well as GBDP-specific MAb. On day 1 post-CCI, the staining patterns with both antibodies in the injured cortex were not different from those in the contralateral cortex (Figure 7A). However, GFAP and GBDP staining of the ipsilateral cortex are greatly increased by D3 and D7 post-CCI. In addition, there is a strong overlap between GFAP core Ab and GBDP-specific antibodies on both D3 and D7 post-CCI. In contrast, there were minimal detections of GBDP in the sham groups (D1 and D7). (Figure 7A). Thus, in the ipsilateral cortex where the cortical impact occurs, the GFAP and GBDP staining are rather similar, reflecting a significant population of astrocytes is injured in this highly vulnerable brain region (Figure 7A). Quantification of immunoreactive cells shows that both the ipsilateral and contralateral cortex has elevated GFAP and GBDP levels (Figure 7D). We confirm that there was minimal staining of GBDP in the control (sham) cortex (Figure 7D).

We then examined by IHC the total GFAP and GBDP staining patterns in the hippocampus. In the CCI model, the hippocampus was not directly impacted, but rather compressed by the contusive force transduced through the impact on the cortex above it. CA1 and CA3 cell layers and the dentate gyrus (DG) in the Injured hippocampus at 7 days post-CCI were examined with total GFAP and anti-GBDP antibodies (Figure 7B,C). Importantly, Figure 7B,C show a significant difference between total GFAP antibody and GBDP-specific antibody staining in hippocampal CA1 and CA3 and DG regions (Ipsilateral side) in CCI samples. Anti-GFAP detected many more cells, while GBDP detect only a fraction of those cells. The differences are also apparent on the contralateral side for CCI (Figure 7B,C). On the contrary, in the sham ipsilateral hippocampus, while baseline GFAP staining was found, there was minimal GBDP staining in CA1, CA3 and DG regions. These observations were confirmed by cell counting shown in Figure 7D. Moreover, we noticed that morphologically they also appear different. While the anti-total GFAP antibody often stains astrocytes in the hippocampus with extended glia processes, GBDP-stained cells show almost no extended processes or many abbreviated processes—consistent with an astroglial injury phenotype (Figure 7B,C). We again noted that in the quantification data (Figure 7D), while GFAP was detected in the sham hippocampus, GBDP was minimally detected in the sham hippocampus (CA1, CA3, and DG). Taken together, the IHC results showed that a subpopulation of injured astrocytes can be uniquely detected with GBDP-specific Ab in the hippocampus and cortex at day 7 post-injury.

### 2.7. Intact GFAP and GBDP Release into Human CSF after TBI

We then examined the translational potential of GBDP formation in human TBI conditions.

We used CSF samples collected within the first 24 h post-injury from severe TBI subjects (n = 21) from a previous study (see Appendix A for patient demographics and clinical data) and compared them to normal control CSF (n = 16). Immunoblots were employed with GFAP-core antibodies. The rationale of this approach is that it can simultaneously monitor both intact and major GBDPs. In fact, with the majority of TBI CSF samples, we observed both intact GFAP and the GBDP-38K, which appear to be the dominant GBDP species (Figure 8A). As a control, we also observed elevated SBDP150 levels in TBI CSF samples over control CSF samples. Quantification shows that the median levels of GBDP-38K in human acute TBI CSF samples are higher than those in normal control CSF (Figure 8B). In parallel, we also observed that intact GFAP was also elevated in human TBI CSF samples. Receiver operating characteristic curve (ROC) for distinguishing TBI versus control CSF showed an AUC of 0.944 for GBDP-38K, which is higher than that for intact GFAP (AUC 0.909) (Figure 8C,D).

### 2.8. Release of Low Molecule Weight GFAP Peptides into Human CSF after TBI

We also extended the proteolytic peptide concept to human TBI CSF (n = 10) versus healthy control CSF (n = 2) samples. They are first subjected to ultrafiltration (5K or 10K MWCO) and the ultrafiltration was then analyzed by unbiased or targeted LC/MS/MS. We identified a number of N- and C-terminal GFAP peptides in TBI CSF ultrafiltrates, but not in the control CSF counterparts. Tentative assignments for the N-terminal peptides span aa 9–49, while C-terminal peptides span aa 386–407 and 417–432. For example, Figure 9 shows matching peptide sequence assignments for MS/MS spectra from unbiased LC/MS/MS, from both de novo sequencing (panel A, B) and database searching (panel C, D), confirming the identity of two of the C-terminal GFAP peptides (DGEVIKES and DGEVIKE) spanning aa 417–424 found in pooled human TBI CSF samples (n = 10 at 6–12 h post-injury), but not in pooled control CSF (n = 10, data not shown). This was further confirmed with targeted LC/MS/MS as shown in Figure 10. Briefly, we synthesized heavy isotope-labeled DGEVIKES and DGEVIKE for validation with an independent set of pooled samples from a different cohort of subjects: healthy control CSF (n = 14), CSF from TBI subjects was collected from the same patients within 6–12 h post-admission (acute n = 15) or 96–120 h post-admission (subacute, n = 13). Figure 10A,B depicts light and heavy peptide signals from both DGEVIKES and DGEVIKE in each of the three pooled samples (see Appendix A for parameters). Based on the ratio of the native light vs. heavy isotope-labeled peptide, we were able to quantify both GFAP C-terminal DGEVIKES and DGEVIKE and established that they were significantly elevated in individual non-pooled TBI CSF samples versus non-pooled healthy controls CSF samples.

Taking all the human CSF data together, in addition to intact GFAP, both the GBDP-38K as well as unique GFAP-released C-terminal proteolytic peptides are present in CSF following TBI. Thus, these GFAP species might have potential as TBI disease progression tracking biomarkers.

## 3. Discussion

In this study, we comprehensively characterized the formation of GFAP breakdown products due to cell-injury/death-linked protease activation. First, GFAP is truncated at both the C- and N-terminals by cytosolic protease calpain to GFAP breakdown products (GBDP) of 46–40K then 38K following pro-necrotic (A23187) and pro-apoptotic (staurosporine) challenges to primary rat astroglia culture or neuron-glia co-culture (Figure 1 and Figure 2).

On the other hand, with the pro-apoptotic challenge (EDTA) where only caspases are activated but not calpain, GFAP was fragmented internally at two sites, generating an N-terminal GBDP of 22K and C-terminal GBDP of 20K (Figure 1 and Figure 2). Purified GFAP protein treatment with calpain-1 and -2 revealed (i) two major N-terminal cleavage sites at L55-A56*N60-A61, and (ii) major C-terminal cleavage sites at T-383*Q-388, producing a limit fragment of 38K and currently releasing several small N-terminal and C-terminal GFAP peptides. (Figure 3, Appendix A).

We have also demonstrated that GBDPs are potential biofluid-based biomarkers for astroglia injury. First, in cultured primary rat astroglia cells challenged with various cytotoxins (A23187, STS, EDTA), we observed robust release of several GBDPs into cell-conditioned culture media, paralleling levels of astroglia cell death measured by LDH release (Figure 2). Lastly, as clinical translation, we observed that CSF samples collected within 24 h after severe TBI in humans show elevated levels of calpain-generated GBDP-38K (Figure 8).

Some early literature has suggested the vulnerability of GFAP to proteolysis. For example, Oh et al. showed that subjecting cultured rat primary astrocytes to acidic pH increases immunoreactivity of GFAP [42]. The same group further reported concurrent fragmentation of GFAP with acidic pH treatment of astrocytes. In addition, both acidic pH-mediated increased immunoreactivity and fragmentation of GFAP were blocked by Ca2+ channel antagonists, suggesting the involvement of calpain [29]. In parallel, Fujita et al. [30] reported increased fragmented GFAP (45K, 37K, and 36K) levels in the spinal cord tissue in patients with amyotrophic lateral sclerosis (ALS). They suggested some cleavage sites at the N-terminal. Such fragmentation pattern is consistent with our finding of calpain-mediated GFAP truncation. Chen et al. previously reported that GFAP might be sensitive to caspase-6 with a cleavage site at D225*V226. We observed this cleavage site with caspase-6 (Figure 2 and Figure 3). But we also observed an additional cleavage site at ELND78*R79 that produced the larger caspase-6 fragment of (GBDP44K) (Figure 1, Figure 2 and Figure 3). In addition, we reported that rather expectedly GFAP is relatively resistant to proteolysis of caspase-3 (Appendix A), as caspase-3 is generally regarded be the most dominant apoptotic cell death execution protease.

It is important to point out that although GFAP is vulnerable to both calpains and caspase-6 fragmentation under in vitro conditions, under acute brain injury conditions such as TBI in mice and human TBI, we observed predominant calpain-mediated signature GBDP-38K (Figure 8 and Figure 10). In addition, using the calpain-generated new N-terminal-directed GBDP-specific antibody, we indeed found strong evidence of the widespread presence of calpain-mediated GBDPs in the cortex and hippocampus after TBI in mice (Figure 8). Thus calpain-mediated GFAP fragmentation is likely the more dominant event under acute brain injury conditions. We also previously shown that when a calpain inhibitor SNJ1945 was given I.P. in vivo to a model of repeated close head injury, it was able to attenuate GBDP44K-38K at day 3, before eventually losing its inhibition by day 7 likely due to instability of the peptidic calpain inhibitor in circulation and the brain [43].

We recently published a preliminary report showing that calpain-generated GBDPs 44–38K were also observed in spinal cord tissue in rats after experimental spinal cord injury (SCI), as well as in CSF during the acute phase after human SCI [44]. Astrocyte injury and death have also been reported to be a feature in other neuronal injury conditions such as focal ischemia in rats [45]. The presence of 36–44 kDa GFAP breakdown products has also been noted in the Alzheimer ‘s brain based on two independent proteomic studies [46,47], which further raises the possibility of using GBDP as an astroglial cell injury marker in chronic neurodegenerative conditions as well.

Thus, taken together, others’ findings and our new data suggest that GBDP can potentially be a multi-purpose biomarker of astroglia injury. For example, GBDP as astroglia injury markers can be examined with immunoblotting, or immunocytochemistry/immunohistochemistry with the use of GBDP N-terminal specific antibodies, under in vitro (in culture) or in vivo (animal studies) conditions. Similarly, the level of glial cell injury in human acute CNS injury or chronic neurodegenerative conditions could be monitored with quantitative immunoblotting (e.g., CSF) or potentially with sandwich ELISA specific to GBDP (blood). Lastly, our observation is that in injured astroglia cells, GFAP might release small GFAP-release C-terminal and N-terminal peptides into biofluid (CSF or blood) that are detectable by mass spectrometry or possible ELISA method.

In terms of other GFAP post-translational modifications, previously GFAP is also known to be regulated by phosphorylation at multiple sites, thus potentially leading to filament disassembly [48,49,50]. Site-specific phosphorylation (Ser13) has recently been identified to be a better marker of Alexander disease severity than total GFAP alone [51]. In our current studies, we focused on acute astroglial injury or acute traumatic brain injury in vivo. We have not investigated GFAP potential phosphorylation as it is beyond the scope of our current study design. But GFAP phosphorylation regulation is certainly an interesting mechanism and we intend to be examined this in future studies.

There are several limitations of the current study. (i) Since this is a pilot verification study of post-TBI GBDP formation and release, we did not focus on having large enough a sample size to generate clinical results correlation (e.g., how the levels of GBDP in CSF correlate with GCS, CT, or MRI neuropathology outcome, (ii) it will be advantageous to quantify GBDPs by developing GBDP-specific quantitative ELISA—this will enable more quantitative and sensitive assessment of GBDP levels in a wider range of biofluids (such as serum and plasma) [52]. It is also noteworthy to assess the correlation between GBDP and blood flowing after TBI or other neurological disorders. As mentioned above, Battaglia et al. recently used Alexander disease (AxD) patient brain tissue and induced pluripotent stem cell (iPSC)-derived astrocytes to identify that AxD-causing mutations lead to selective phosphorylation of GFAP at Ser13 and subsequent GFAP cleavage by caspase-6 and cytoplasmic aggregation formation in patients who died young [51]. Thus, caspase-6 cleavages of GFAP might be important in defining chronic neurodegenerative conditions. In contrast to our study here, we presented clear evidence that GBDP-38K is only produced during acute astroglial injury events but not in normal astrocytes in the culture at present, (Figure 2 and Figure 4C), and animal model of TBI (Figure 5). However, we do not yet have sufficient evidence to demonstrate if measuring the calpain-generated GBDP-38K is superior to measuring total GFAP from a brain injury diagnosis or prognosis standpoint. Once GBDP-specific ELISA is developed we should be able to address these important questions.

As for the GFAP DGEVIKE and DGEVIKES peptides (Figure 8), the calculated concentration of the endogenous peptides in this experiment helped to provide proof-of-concept data that the endogenous peptides can be detected in TBI CSF samples that were processed with the same experimental protocol as the discovery TBI samples. We found that this protocol would need to be further optimized to improve the detection of the GFAP peptides before LC-SRM-MS detection, but these current data provide a starting point for the development of a robust and validated method for analysis of future biosamples.

Promising findings from animal studies led to the assessment of the clinical significance of GFAP in patients suffering from severe and mild to moderate TBI. The elevation of GFAP in biofluids was associated with injury severity and clinical outcomes [28].

In summary, our study suggests that by harnessing the various forms of GFAP breakdown products as biomarkers, one can gain mechanistic insights and tracking of astroglia injury under various neural injury or neurodegenerative conditions where calpains and/or caspase-6 are activated [25].

## 4. Materials and Methods

All animal protocols conform to the National Institutes of Health guidelines on the use of laboratory animals and are approved by the University of Florida Institutional Animal Care and Use Committee (#201207692, 201207683, 201207558, 201308116).

### 4.1. Primary Rat Cortical Mixed Neuron-Astroglia Culture, Primary Rat Astroglia Culture, and Treatment

Primary cortical mixed neuron-astroglia cultures were prepared from embryonic day 18 Sprague–Dawley rat fetuses and plated on poly-L-lysine coated 12-well plates (Erie Scientific, Portsmouth, NH, USA). Cells were allowed to grow in an atmosphere of 10% CO2 at 37 °C for 3 days and then treated with 1 mM 4-amino-6-hydrazino-7-D-ribofuranosyl-7H-pyrrole (2,3-D)-pyrimidine-5-carboxamide (ARC) for 2 days. The ARC was removed, and fresh 10% plasma-derived horse serum (PDHS) in DMEM was added, after which the cells were grown for an additional 10 to 14 days in 12-well plates. For cytotoxic challenges, in addition to untreated controls, the following conditions were used: calcium ionophore A23187 (25 μM; Sigma, St. Louis, MO, USA) as a calpain-dominated pro-necrosis challenge for 24 h; staurosporine (STS; 2 μM; Sigma) as caspase/calpain-dependent apoptotic inducer and 5 mM EDTA as a calpain-independent apoptotic inducer for 24 h. For activation of astroglial cells, 100 ng/mL of LPS (lipopolysaccharide, from *E. Coli*, Sigma) was used. For pharmacologic intervention, cultures were pretreated 1 h before the A23187 or STS, EDTA or LPS challenge with either the calpain inhibitor SNJ-1945 (a gift from Senju Pharmaceuticals, Kobe, Japan) or the pan-caspase inhibitor Z-D-DCB or caspase-6 inhibitor Z-VEID-FMK (R&D, Minneapolis, MN, USA).

### 4.2. In Vivo Model of Traumatic Brain Injury-Controlled Cortical Impact in Mice

A controlled cortical impact (CCI) device was used to model TBI as in the previous report. Briefly, after reaching a deep plane of anesthesia, CB57BL/6 mice (male, 3 to 4 months old, Charles River Laboratories, Raleigh, NC, USA) were mounted in a stereotactic frame in a prone position. A midline cranial incision was made and a unilateral (ipsilateral) craniotomy (4 mm diameter) was performed adjacent to the central suture, midway between the bregma and the lambda. Brain trauma was induced using an Impact One stereotaxic impactor (Leica Biosystems Inc. Lincolshire IL, USA) by impacting the right cortex (ipsilateral cortex) with a 3 mm diameter impactor tip at a velocity of 3.5 m/second, 1.5 mm compression depth, and a 200 ms dwell time (compression duration). Sham-injured control animals underwent identical surgical procedures but did not receive an impact injury. Naïve animals underwent no procedure.

### 4.3. Cell Lysate/Brain Tissue Collection and Preparation

For immunoblot analysis, primary neuronal cells were collected and lysed for 90 min at 4 °C with a lysis buffer containing 50 mM Tris (pH 7.4), 5 mmol/L EDTA, 1% (*v*/*v*) Triton X-100, 1 mmol/L DTT, and a Mini-Complete protease inhibitor cocktail tablet (Roche Biochemicals). Injured cortical and hippocampus tissues in TBI animals and cortex in similar areas from naïve animals were isolated after mice reaching deep anesthetized and were euthanized by decapitation. The tissue samples were pulverized to a fine powder with a small mortar and pestle set over dry ice. The pulverized brain tissue was then lysed for 90 min at 4 °C with lysis buffer as described above. Cell or tissue lysates were then 10,000× *g* for 10 min at 4 °C. The supernatants were snap-frozen and stored at −80 °C until use.

### 4.4. In Vitro Calpain or Caspase Digestion of Rat Cortex Lysate or Purified or Purified GFAP Protein

Rat cortex lysate was obtained by incubating with lysis buffer containing 50 mM Tris–HCl (pH 7.4), 5 mM LEDTA, 5 mM EGTA, 1% Triton X-100, and 1 mM DTT for 90 min at 4C. In vitro digestion of cortex lysate (100 μg) or purified recombinant human GFAP protein (3 μg, DxSys, Mountain View, CA, USA) was performed with human erythrocyte calpain-1 (EMD Bioscience, Billerica, MA, USA) and human recombinant caspase-3 (BD Pharmingen, San Jose, CA, USA) or recombinant caspase-6 (Enzo Life Tech.) in a buffer containing 100 mM Tris–HCl (pH 7.4), 20 mM DTT, with or without 1 mM CaCl2, at room temperature for 30 min (calpain-1) at protease to substrate ratio of 1/100 (unless stated otherwise), or 4 h with caspase-6 or caspase-3 at protease: substrate ratio of 1/25 (unless stated otherwise). Protease reactions were stopped by the addition of a protease inhibitor cocktail solution (Roche Biochemicals, Indianapolis, IN, USA).

### 4.5. Lactate Dehydrogenase Release Assay of Cell Death

To evaluate the cytotoxicity, a lactate dehydrogenase (LDH) release assay (Cyto-Tox One Reagent, Promega, Madison, WI, USA) was performed. Primary neuron-astroglia mixed culture or astroglia cells were cultured as described above and then treated with undigested or digested recombinant GFAP protein with either calpain or caspase-6. Culture media were collected 24 h after treatment and assayed for LDH release by following the manufacturer’s instructions. Briefly, 50 μL of the reconstituted 2X LDH assay buffer was added to the 50 μL supernatant. Incubate the plate was incubated at room temperature (22–25 °C) for 10–30 min. Then 50 μL of stop solution was added to the wells. The absorbance was measured at 490 nm. Three to four replicates were assayed.

### 4.6. Human Cerebrospinal Fluid Samples

The CSF samples from a severe TBI study were collected from consented adult subjects presenting to the Emergency Department of Ben Taub General Hospital, Baylor College of Medicine (Houston, TX, USA). The study protocol was approved by the Baylor College of Medicine IRB. The study abided by the Declaration of Helsinki principles. TBI subjects with a Glasgow coma scale of 3–8 were enrolled and the CSF samples were collected as described in the previous study [41] according to the hospital’s standard procedures. Timed CSF samples were collected within 24 h of a head injury and underwent an immediate sample preparation as follows. The 5 to 10 mL CSF was 4000× *g* at room temperature for 5 min to remove loose cells and debris. The cleared CSF (supernatant) samples were aliquoted into 1-mL vials followed by snap-frozen and stored at −80 °C ultralow freezer until use. About 10 μL of each CSF sample was used for immunoblot analysis. The control CSF samples were purchased from Bioreclamation (Westbury, NY, USA) [41].

### 4.7. Immunocytochemistry/Immunohistochemistry

Immunocytochemistry analysis was performed on rat primary astroglia cultures. Cells were fixed with 4% paraformaldehyde for 10 min, washed with PBS, and permeabilized with 0.1% Triton X-100 for 5 min. Routine staining was performed after a 1-h blocking step in 2% goat serum. Immunohistochemistry analysis was performed on 4 to 6 μm frozen brain tissue sections. After perfusion with 4% paraformaldehyde, dehydration in 30% sucrose, and sectioning, brain slides were blocked with 2% normal goat serum, followed by a routine staining procedure. Monoclonal mouse-anti-GFAP (#556330, BD Biosciences Int. CA, USA) and polyclonal rabbit anti-GBDP (custom-made at Univ. Florida) antibodies at a dilution of 1:200 were used. Alexa 488- or Alexa 568-conjugated goat–anti-rabbit secondary antibody (Molecular Probes, Eugene, OR, USA) was added at a dilution of 1:1000. The cells/tissues were counterstained with 4,6-diamidine-2-phenylindole (Vector Laboratories, Burlingame, CA, USA). Fluorescent images were captured with a 40× objective on the OLYMPUS DP71 fluorescent microscope (Olympus America Inc., Center Valley, PA, USA).

### 4.8. SDS–PAGE, Electrotransfer and Immunoblot Analysis

Protein concentrations of cell or tissue lysates were determined using Bio-Rad DC Protein Assay kits (Bio-Rad, Hercules, CA, USA). Protein-balanced samples were prepared in a sample loading buffer containing 0.25 mM Tris (pH 6.8), 0.2 mM DTT, 8% SDS, 0.02% bromophenol blue, and 20% glycerol in distilled water. Immunoblot analysis was performed as previously described [41]. Briefly, the same amount of protein from each sample (20 µg) was loaded for SDS-PAGE electrophoresis. Subsequently, immunoblotting was performed following standard procedures with the following antibodies: GFAP core Mab (556330; BD Bioscience), GFAP C-terminal Pab (48050, Abcam), GBDP-specific PAb (custom-made at Univ. Florida) and alpha-fodrin (αII-spectrin) (1:1000, BML-FG6090, ENZO Life Sciences Farmingdale, NY, USA). Protein detection was further visualized using biotin, avidin-conjugated alkaline phosphatase, nitro blue tetrazolium, and 5-bromo-4-chloro- 3-indolyl phosphate. A 250 K to 14 K rainbow molecular weight marker (RPN800E, GE Healthcare, Bio-Sciences, Pittsburgh, PA, USA) was used to identify the protein. Quantitative evaluation of protein levels was performed via computer-assisted densitometric scanning (NIH ImageJ, version 1.6 software).

### 4.9. CSF Samples Processing for Ultrafiltration

Biosamples (100 uL) are subjected to ultrafiltration (250 μL unit, with 5000 or 10,000 Da molecular weight cut-off (MWCO) membrane filter (Sartorius Stedim Biotech, Goettingen, Germany). The sample-loaded unit was centrifuged at 4 °C for 5–20 min and the filtrate collected is then concentrated five-fold. Concentrated retentate containing large protein fragments or undigested protein is routinely run on SDS-PAGE and Western blot to confirm proper protein digestion. Filtrate for subjected to LC/MS/MS peptide analysis.

### 4.10. Unbiased Liquid Chromatography-Tandem Mass Spectrometry (LC/MS/MS)

Sample preparation: Fifty microliters of each CSF sample were filtered with a 10 kDa molecular weight membrane filter at 12K× *g* for 25 min. The flow-through was dried by vacuum centrifugation and dissolved in 10 μL of 2% acetonitrile/0.1% formic acid in water. Three microliters were injected into each sample for unbiased LC/MS/MS. Analysis: Nanoflow was performed on a NanoAcquity UPLC (Waters, Milford, MA, USA); the autosampler was used to load two microliters of each concentrated sample onto a 5 μm particle size Symmetry 180 μm × 20 mm C18 trapping column at 4 µL/min for 10 min. Then, the sample plug was loaded onto a 1.7 µM particle size BEH130 C18 100 µm × 100 mm analytical column at 300 nL/min. The mobile phase consisted of solvent A (water with 0.1% formic acid) and solvent B (acetonitrile with 0.1% formic acid). Separation was achieved within a run time of 170 min at a flow rate of 300 nL/min. The first linear gradient was from 1% to 40% B over 145 min, and the second linear gradient was from 40% to 100% B over 5 min and held for 5 min before returning to initial mobile-phase composition (1% B). MS/MS spectra were collected on a FT-ICR/Orbitrap (linear ion trap) with nanospray (Thermo) using a Data Dependent Acquisition method (Thermo), in which data-dependent scanning was specified as a criterion to select the top 10 most abundant ions using 11 separate scan events at a given chromatographic time point (115 min) for subsequent analysis. The mass spectrometer was set to perform a full scan and subsequently, MS/MS scans on the ten most intense ions in the full-scan spectrum MS (scan event 1) with Dynamic Exclusion enabled. Dynamic Exclusion temporarily puts a mass into an exclusion list after its MS/MS spectrum is acquired, providing the opportunity to collect MS/MS information on the second, third, etc. most intense ion from the full-scan spectrum MS (scan event 1). All 36,954 MS/MS spectra from 12 files (10 hTBI + 2 human controls) were analyzed with PEAKS PRO (46–50) for de novo sequencing and database searching of 20,395 reviewed human protein sequences in the UniProtKB (i.e., Swiss-Prot). The search was achieved using the average mass for matching the precursor with a fragment ion mass error tolerance of 0.1 Da and a precursor ion mass error tolerance of 0.2 Da. Carbamidomethylation of cysteine was selected as a static modification, while the oxidation of methionine was selected as a dynamic modification. De novo sequencing: 11,563 de novo peptides were remaining after a de novo score filter ≥ 50%. Identified de novo peptides were exported to MGF files. Database searching: Exactly 8171 peptide-spectrum matches (PSMs) were identified with a 1.0 false discovery rate (FDR) at a −10lgP score ≥ 38.1. Identified PSMs were exported to MGF files.

### 4.11. Targeted LC/MS/MS (a.k.a.LC-SRM-MS)

Materials: Acetonitrile water and formic acid were LC/MS Optima grade and purchased from Thermo Fisher Scientific (San Jose, CA, USA). Synthetic peptides were purchased from 21st Century Biochemical, Inc (Marlboro, MA, USA) with 13C/15N amino acids incorporated into the peptide sequences. GFAP peptides: DGE(V-13C/15N) IKES-OH and DGE(V-13C/15N) IKE-OH. Peptides were synthesized in 1 mg quantities and purified to 98% and supplied at 5 pmol/μL in 30% acetonitrile/0.1% formic acid in water. Amino acid analysis data was provided by 21st Century Biochemical, Inc for peptide concentration determination. Sartorius Vivacon 10 kDa molecular weight filters were purchased from Fisher Scientific. Sample preparation: Fifty microliters of each CSF sample were combined with a 10 μL spike of heavy peptides (20 fmol/μL for GFAP peptides) and filtered with a 10 kDa molecular weight membrane filter at 12K× *g* for 25 min. The flow-through was dried by vacuum centrifugation and dissolved in 10 μL of 2% acetonitrile/0.1% formic acid in water. Three microliters were injected into each sample for targeted LC/MS/MS (selected reaction monitoring). Analysis: Targeted LC/MSMS analysis was developed on a U-3000 RSLCnano with a loading pump and a TSQ Altis triple quadrupole MS using a nano flex ion source (Thermo Fisher Scientific, San Jose, CA, USA). LC separation was achieved using an Acclaim PepMap 100 C18 trap column (0.1 mm × 2 cm, 5 um beads) and an in-house-packed nanocolumn (75 um ID × 15 cm, Dr. Maisch Reprosil AQ C18, 1.9 um beads) with integrated, laser pulled tip (~7 μm ID). Samples were loaded at 4 μL/min with 2% B for 5 min using the loading pump. A nanoflow gradient was then delivered to the trap and resolving column at 200 nL/min according to the following: 2–35%B over 21 min; 35–95% B over 1 min, hold at 95% B for 4 min, 95–2% B over 1 min, and re-equilibration at 2% B for 18 min. Mobile phase A was 0.1% formic acid in water (*v*/*v*) and mobile phase B was 0.1% formic acid in acetonitrile (*v*/*v*). The nanospray voltage was 1600 V and applied between the HPLC and the nanocolumn using a low volume tee. The ion transfer tube was heated to 325 degrees C. Peptide precursor charge states and product ions were selected by targeting the predicted Q1/Q3 transitions (Skyline, University of Washington) and refining the transition list to monitor a minimum of 6 products ions per peptide. The collision energy was optimized at the transition level using the purified synthetic isotopically labeled peptides via LC/MS/MS [53] where more details are described in Supplemental Data. The MS cycle time was set to 0.8 s, with dwell times of 9.6 msec per transition (72 transitions monitored per cycle) and Q1 and Q3 resolution was set to 0.7 FWHM. The CID gas pressure was 1.5 mTorr and the “Use Chromatographic Filter” option was enabled. Optimization: Isotopically labeled peptide standard spike levels were optimized on pooled CSF samples. Ten microliters of each TBI subtype (healthy, acute, and subacute) were pooled to generate ~150 uL subtype pools. Fifty microliters of each subtype pool were processed as described above to determine optimal heavy peptide concentrations for detection. Skyline software [53] was used for data analysis.

### 4.12. Statistical Analysis

In vitro cell treatment, animal tissue lysate, and protein digestion experiments were performed in triplicate per experiment, and mean values are used, and the mean values from three independent experiments performed were used to create a group mean based on the total sample size. For experiments concerning in vivo TBI models, the minimum number of animals per group at each time point was five (n = 5). For samples from humans, 15 control and 21 TBI subjects were included in the CSF analysis. Data are shown for GFAP protein and GBDP-38K median (interquartile range). Statistical significance was determined with median comparison by the Kruskal–Wallis test for the groups, (*p* values < 0.05 for both peptides). We also conducted Dunn’s comparison test between acute and subacute TBI to control. respectively (* *p* < 0.05)

## Figures and Tables

**Figure 1 ijms-23-08960-f001:**
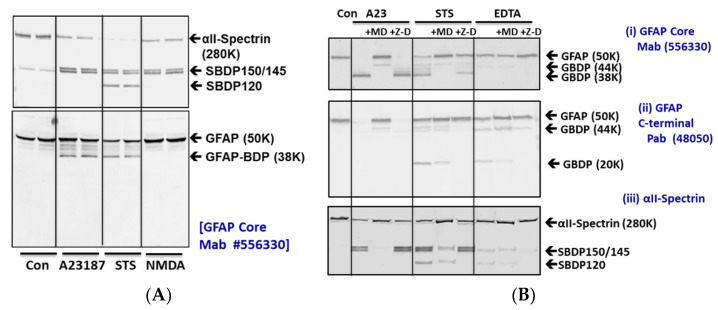
GFAP proteolysis versus αII-spectrin breakdown in cerebrocortical neuron-astroglial mixed culture (CTX) after subjecting to cytotoxin challenges (24 h). (**A**) CTX was subjected to A23187, STS, or NMDA treatment for 6 h. Cell lysate probed with αII-Spectrin or GFAP core Mb. (**B**) CTX treated with A23187, STS, and EDTA for 24 h in the absence or presence of calpain inhibitor MDL-28170 or pan-caspase inhibitor Z-D-DCB. Cell lysate probed with αII-Spectrin or GFAP core MAb or GFAP C-terminal PAb. Blots are representative of four separate experiments. CTX subjected to STS treatment in the absence or presence of caspase-6 inhibitor Z-IVED-FMK or with calpain inhibitor MDL-28170.

**Figure 2 ijms-23-08960-f002:**
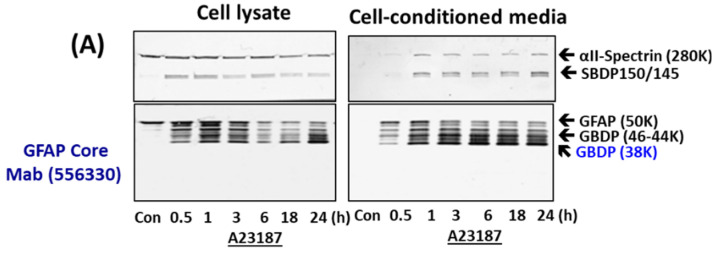
GBDP formation in astroglial cells and release into cell-conditioned media parallel cell injury in primary astroglial culture challenges with pro-necrotic (A23187) and pro-apoptotic cytotoxins (STS, EDTA). (**A**) Cell lysate or cell media probed with αII-Spectrin, GFAP core MAb or GFAP C-terminal PAb. (**B**,**C**) Quantification of GBDP formation and release parallel astroglial cell injury measured by LDH release with a time course of A23187, STS and EDTA treatment. Panel (**D**) quantification of LDH releases while (**E**,**F**) are GFAP-38K quantification in cell lysate and cell-conditioned media respectively. Shown are mean ± SEM (n = 4). * *p* < 0.05, ** *p* < 0.01 when compared to control (student *t*-test).

**Figure 3 ijms-23-08960-f003:**
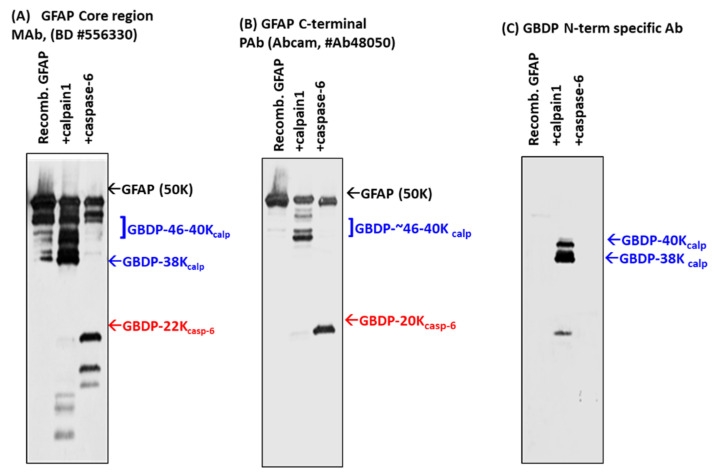
Digestion of purified GFAP by calpain-1 and caspase-6. Characterization with different GFAP antibodies and GBDP-specific antibodies by immunoblotting. (**A**) GFAP core Mab, (**B**) GFAP C-terminal Pab, (**C**) GBDP-specific antibody. (**D**) GFAP proteolysis with calpain, caspase-6, and predicted major fragment sizes. Data based on N-terminal sequencing. (**E**) Schematics of GFAP proteolytic fragment formation. The top panel shows GFAP linear model and major calpain/caspase-6 cleavage sites. The middle panel shows the positions of major fragments. The bottom panel shows the human GFAP amino acid sequence and the annotated cleavage sites (*) by calpain (blue), and by caspase-6 (red).

**Figure 4 ijms-23-08960-f004:**
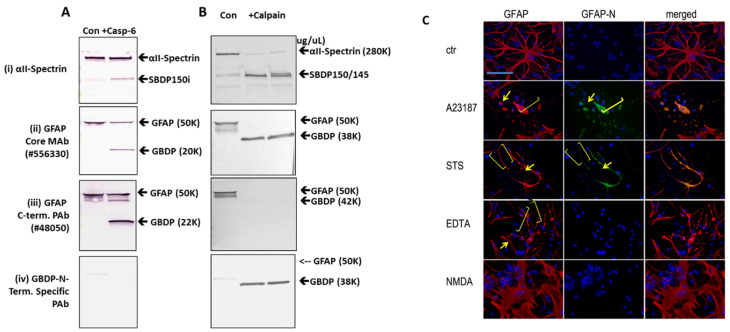
Characterization of calpain-derived GBDP (38K)-N-terminal specific antibody: immunoblotting of in vitro digest of GFAP in cerebrocortical neuron-astroglia mixture lysate with caspase-6 (**A**) vs. calpain-1 (**B**). Characterization of αII-Spectrin, GFAP core, and C-terminal antibodies and calpain-derived GBDP (38K)-N-terminal specific antibody. (**C**) Immunocytochemical characterization of rat primary astroglia cells labeled with total GFAP and GBDP-specific antibodies. Rat primary astrocytes subjected to A23187, STS, EDTA, or NMDA were stained with anti-GFAP core MAb, GBDP-N-terminal specific antibody. Yellow arrows show distinct GBDP-staining of the astrocyte cell body. Yellow brackets show degenerative astrocyte processes.

**Figure 5 ijms-23-08960-f005:**
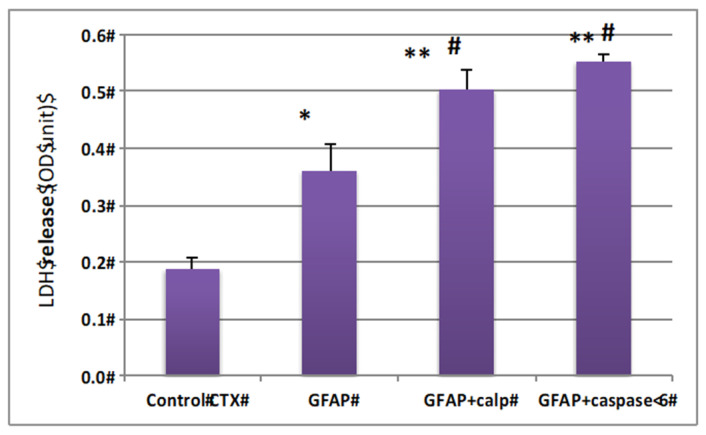
GFAP protein and GBDP are neurotoxic to rat cerebrocortical culture. Cultured cells were treated with 500 ng GFAP fragment (calpain or caspase-6 digestion) * compared with control *p* < 0.05, ** compared with control *p* < 0.01; # compared with intact GFAP treatment, *p* < 0.05.

**Figure 6 ijms-23-08960-f006:**
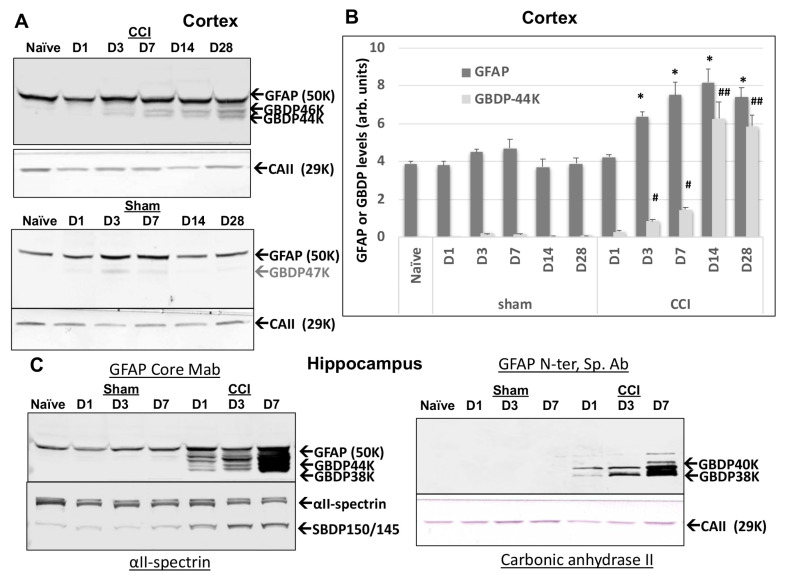
Ipsilateral cortex GFAP and GFAP-breakdown product (GBDP) profile by immunoblotting at different time points after controlled cortical impact (CCI) injury in mice. Time points are D1, 3, 7, 14, or 28 after injury. (**A**) Representative blots for GFAP and housekeeping gene loading control (Carbonic anhydrase II, 29 kDa). (**B**) Quantification of GFAP and GBDP-44K (equalized by CA-II band intensity). * *p* < 0.05, when compared with naïve GFAP, # *p* < 0.05; ##, *p* < 0.01, when compared with naïve GBDP-44K. (**C**) GBDP-specific MAb did not detect naïve or sham hippocampus but sensitively detected the formation of GBDP-40K and GBDP-38K at D3 and D7 post- CCI samples.

**Figure 7 ijms-23-08960-f007:**
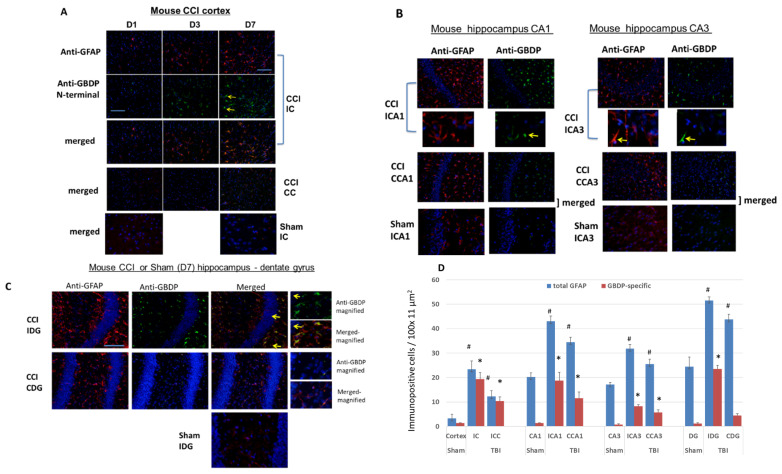
Immunohistochemical staining with an injured mouse brain with total GFAP and anti-GBDP N-terminal specific antibodies. CCI mouse. Red: anti-GFAP, green: anti-GBDP. Shown are injured cortex (**A**), injured hippocampus [CA1 and CA3 cell layers (**B**) and dentate gyrus (**C**)] with total GFAP and anti-GBDP N-terminal specific antibody. 7 days post-CCI mouse brains are examined. Red: anti-GFAP, green: anti-GBDP. ICA1, ipsilateral CA1, CCA1, contralateral CA1, ICA3, Ipsilateral CA3, CCA3, contralateral CA3. Yellow arrows indicated GBDP Antibody labeled cells. (**D**) quantification of total GFA or GBDP positive cells from control rats (naïve) or rats 7 days after CCI, ipsilateral and contralateral cortex, and hippocampus (CA1, CA3, and dentate gyrus (DG)) were used. Shown are mean ± SEM (n = 4). * *p* < 0.05, ^#^ *p* < 0.05 when compared to control. (student’s *t*-test).

**Figure 8 ijms-23-08960-f008:**
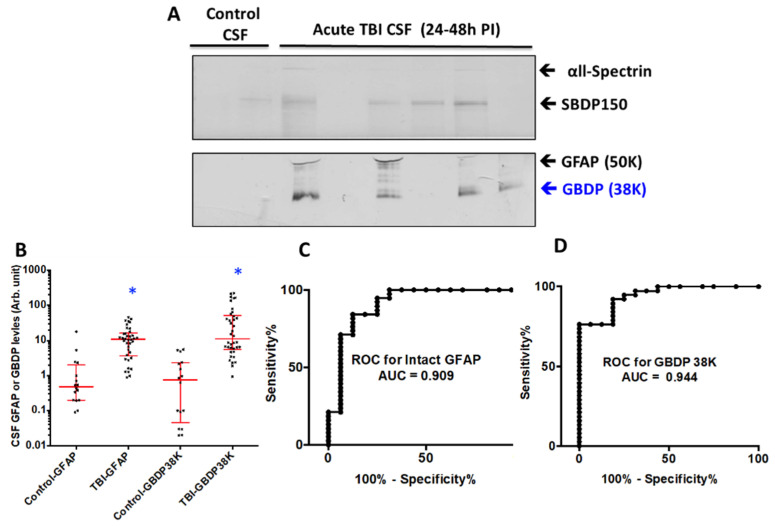
GFAP and 38K GBDP in human CSF samples within 24 h of severe TBI. (**A**) Representative blots showed αll-Spectrin fragment SBDP150, intact GFAP and GBDP-38K in human control and TBI CSF samples. (**B**) Scattered plot for control and TBI CSF (collected within 24 h post-admission). Groups are shown as the median and interquartile range (IQR), * *p* < 0.05. Control has N = 12, TBI N = 30. (**C**,**D**) ROC showed higher AUC for GFAP-38K (0.944) that intact GFAP (0.909).

**Figure 9 ijms-23-08960-f009:**
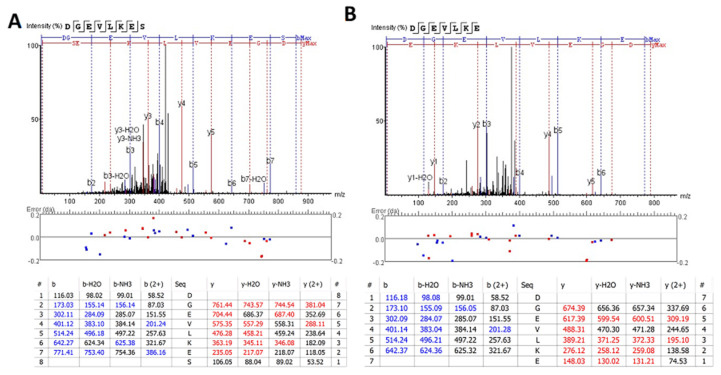
Unbiased LC/MS/MS followed by (**A**,**B**) de novo sequencing or (**C**,**D**) database searching both revealed GFAP peptides from pooled human acute TBI CSF samples from n = 10 samples (6–12 h post-injury): (**A**,**C**) DGEVIKES and (**B**,**D**) DGEVIKE as shown by the annotated spectra. Note: I/L are isobaric, based on the amino acid sequence for human GFAP DGEVLKES and DGEVLKE (de novo sequencing) &DGEVIKES and DGEVIKE (database searching).

**Figure 10 ijms-23-08960-f010:**
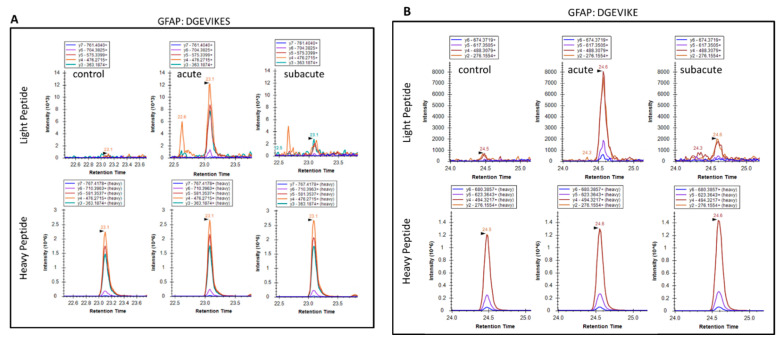
Targeted LC/MS/MS extracted ion chromatograms for light and heavy versions of GFAP peptides in control and in acute and subacute time points: (**A**) DGEVIKES and (**B**) DGEVIKE. Heavy peptides were spiked in at 20 fmol/50 uL of pooled CSF before ultrafiltration with a 10 kDa membrane. Pooled CSF samples were generated by pooling 10 uL of each sample type (control, n = 14; acute TBI, n = 15; subacute TBI, n = 13) from the available CSF samples. The data shown are for a single analysis of each pooled sample type.

## Data Availability

Unbiased LC/MS/MS data and results have been deposited in the MassIVE proteomics data repository [54] (ID: PXD026494, Username: MSV000087573_reviewer, Password: L84dnr!?!), while targeted LC/MS/MS data have been deposited in Panorama Public proteomics data repository [55] (ID: PXD026550, Username: panorama+reviewer34@proteinms.net, Password: vClOyrZJ, URL: https://panoramaweb.org/GB-1.url (accessed on 19 June 2022)).

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
