# Peer review of "Characterization of Calpain and Caspase-6-Generated Glial Fibrillary Acidic Protein Breakdown Products Following Traumatic Brain Injury and Astroglial Cell Injury"

_ijms, 2022, doi:10.3390/ijms23168960_

Round 1

Reviewer 1 Report

The authors investigated the mechanisms of astroglial damage by examining the potential vulnerability of GFAP proteins to proteolytic attack and their respective fragmentation patterns under in vitro digestion to calpains and caspases in different materials, from cell cultures to humans.

The paper is very well thought out from a methodological point of view and the results may be significant despite the limitations pointed out by the authors.

I would only recommend expanding the bibliographic entries with papers that have been published in IJMS itself.

Author Response

We appreciate the insightful comments and made adjustments accordingly.

We have incorporated the following papers that are relevant to our topic from IJMS journal and other related journals.

  • Novel Peptidomic Approach for Identification of Low and High Molecular Weight Tauopathy Peptides Following Calpain Digestion, and Primary Culture Neurotoxic Challenges
  • Diffuse Axonal Injury: Clinical Prognostic Factors, Molecular Experimental Models and the Impact of the Trauma Related Oxidative Stress. An Extensive Review Concerning Milestones and Advances

We appreciate your suggestions.

Reviewer 2 Report

This is a well composed paper for researchers in that very specific field. I believe it is a great experiemental work with limited practical impact yet. The design fits into the molecular research trend very well, however the applicability for clinical use should be more emphasized. 

Author Response

We appreciate your insightful comments and we have incorporated one more paper relating to the clinical significance of GFAP in TBI

  • Thorough overview of ubiquitin C-terminal hydrolase-L1 and glial fibrillary acidic protein as tandem biomarkers recently cleared by US Food and Drug Administration for the evaluation of intracranial injuries among patients with traumatic brain injury

Thank you 

Reviewer 3 Report

This is a novel study examining the proteolysis of the astrocyte protein GFAP, with calpains and caspases and following in vitro and in vivo injury models.  The study has demonstrated that GFAP is subject to proteolysis by calpain and caspase 6, and highlights that these breakdown products may have potential diagnostic applications for tracking the progression of brain injury.

The manuscript is well written the studies well executed and the authors should be commended for their work in generating the novel and interesting findings of the study.

I have provided some comments and suggestions for the authors regarding the manuscript.

Abstract

Line 30: “…calpain-1 and -2 generates (i) major N-terminal cleavage sites between A-56 and A-75 and..”  Figure 3E shows calpain A-56 and calpain A-61??

Line 32: “…Caspase-6 treated GFAP was cleaved at D-78, R-79, D-266 and A-267,..”  Figure 3E shows D-78, R-79, D-225 and V-226??    Also maybe use “..D-78/R-79 and D-225/V-226..”  (check in line 646)

Results

Line 108: can “limit fragment” be explained/defined? Does it really need to be called a limit fragment?  What about the 20K GBDP fragment, isn’t that a limit fragment??

Line 110-112: omit “We found that GFAP is quite vulnerable to proteolysis. The 50K intact GFAP is subsequently cleaved to intermediate GBDP between 48K-40K) and then ultimately to the limit fragment of 38K.” – doesn’t really add anything.

Line 116: define “BDP”; i.e breakdown products (BDPs)

Line 177: change “GFAP-BDPs” to “GBDPs” throughout manuscript

Line 119: remove the bracket “)”.

Line 199 define SBDP.

Line 125 replace “Core” with “core”. Check throughout manuscript.

Line 182: Statement “except that intact GFAP-50K was not observed prominently…” Images would suggest GFAP-50K is prominent.  Image analysis of western blots would be useful here.  Although, probably not essential is that a possibility?

How was loading controlled for in Fig 1 and 2.

Figure 2A: for STS challenge why isn’t GBDP-20K fragment present in lysate?

Fig 3D (Table )/line 346.  “N59*A60”.  Shouldn’t that be N60*A61, according to Figure 3.

Fig 3E/line 355.  Check sequence “*A61AGFKET..”.  Shouldn’t that be A61GFKET..”. 

Line 385: “…A60GFKETRASE based on N-ter-..”  should that be A61GFKETRASE.

Line 390: “…In vitro..” change to “in vitro”.

Line 408/ Fig 4C.  Does Figure show GBDP-specific PAb does not label caspase-dominant apoptotic astrocytes???   Note the PaB antibody labels the C-terminus of GFAP, while the antibody used in Fig 4C is labelling N-terminus of GFAP (GBDP- N-terminal specific antibody).  I’m a little confused here.

Line 428: Could Suppl. Figures S5 and S6 be incorporated into the main section of manuscript??

Line 577:  Figure 7A, were all lanes (24-48h PI) in gel loaded with acute TBI CSF samples; if so why wasn’t GDBP-38 detected in all lanes. Figure 7B would suggest all samples contained GDBP-38.

Line 628, Figure legend; “Statistical analysis is shown in Figure 9”.  Where is Figure 9???

Discussion

First 4 paragraphs of Discussion are really just a re-hash of the results.  Consider shorting considerably.

Line 682 to 684. Be consistent when describing cleavage sites; “cleavage site between D225 and V226” or “cleavage site at ELND78*R79”.  Check throughout manuscript.

Line 715: Is there any evidence indicating GBDPs are released into the blood flowing brain injury, and should the feasibility to routinely measure GBDPs in CSF in the clinic after TBI be discussed?

Author Response

We thank you for the thorough read and your insightful comments. We have carefully read your comments and responded to them individually, indicating how we addressed each comment and describing the amended changes. The revisions have been approved by all authors accordingly. This made our manuscript more concise and better understood.

Abstract

Line 30: “…calpain-1 and -2 generates (i) major N-terminal cleavage sites between A-56 and A-75 and..”  Figure 3E shows calpain A-56 and calpain A-61??

Thank you, we have corrected this.

Line 32: “…Caspase-6 treated GFAP was cleaved at D-78, R-79, D-266 and A-267,..”  Figure 3E shows D-78, R-79, D-225 and V-226??    Also maybe use “..D-78/R-79 and D-225/V-226..”  (check in line 646)

Thank you, we have corrected this.

Line 108: can “limit fragment” be explained/defined? Does it really need to be called a limit fragment?  What about the 20K GBDP fragment, isn’t that a limit fragment??

We removed the limit fragment to avoid confusion

We appreciate this comment

Line 110-112: omit “We found that GFAP is quite vulnerable to proteolysis. The 50K intact GFAP is subsequently cleaved to intermediate GBDP between 48K-40K) and then ultimately to the limit fragment of 38K.” – doesn’t really add anything.

We corrected this.

Line 116: define “BDP”; i.e breakdown products (BDPs)

We added the definition

Line 177: change “GFAP-BDPs” to “GBDPs” throughout manuscript

We corrected this.

Line 119: remove the bracket “)”.

We corrected this.

Line 199 define SBDP.

We added the definition

Line 125 replace “Core” with “core”. Check throughout manuscript.

We have corrected this throughout the manuscript

Line 182: Statement “except that intact GFAP-50K was not observed prominently…” Images would suggest GFAP-50K is prominent.  Image analysis of western blots would be useful here.  Although, probably not essential is that a possibility?

Although we are able to detect the GFAP-50K DDP, in some figures, the presence of this particular BDP is not consistent and depends on the protease activation as we have observed in previous unpublished data (unshown data) compared to the  GFAP-38 kDa and the 20/22 kDa that are shown to be accumulating upon caspase/calpain activation. We hope that would clarify the comment by the respected reviewer.

How was loading controlled for in Fig 1 and 2.

For these two gels, since we used cell cultures, we have relied on staining the gel with Coomassie blue to check for equal loading. We apologize for not showing the data. Nevertheless, we have loaded 20 ug of each sample in the gels assessed by protein assay measurement.

Figure 2A: for STS challenge why isn’t GBDP-20K fragment present in lysate?

Thank you for this remark. We agreed that it is showing in the cell conditioned media but not in the cell lysate. This is suggestive that this particular GBDP 20K can actually leak out of the cell to be detected in the lysate which highlights the importance of that GBDP 20-22K as a secreted marker of GFAP proteolysis.

Fig 3D (Table )/line 346.  “N59*A60”.  Shouldn’t that be N60*A61, according to Figure 3.

We corrected this.

Fig 3E/line 355.  Check sequence “*A61AGFKET..”.  Shouldn’t that be A61GFKET..”. 

Line 385: “…A60GFKETRASE based on N-ter-..”  should that be A61GFKETRASE.

Thank you for noticing

We have corrected these.

Line 390: “…In vitro..” change to “in vitro”.

We corrected this.

Line 408/ Fig 4C.  Does Figure show GBDP-specific PAb does not label caspase-dominant apoptotic astrocytes???   Note the PaB antibody labels the C-terminus of GFAP, while the antibody used in Fig 4C is labelling N-terminus of GFAP (GBDP- N-terminal specific antibody).  I’m a little confused here.

In this figure, we are showing that this is highly specific to calpain-dominant apoptotic astrocytes rather than caspase-specific injury (EDTA)

We apologize for the confusion

Line 428: Could Suppl. Figures S5 and S6 be incorporated into the main section of manuscript??

We added the figures and relabeled all the figures accordingly

Thank you for this valuable advice.

Line 577:  Figure 7A, were all lanes (24-48h PI) in gel loaded with acute TBI CSF samples; if so why wasn’t GDBP-38 detected in all lanes. Figure 7B would suggest all samples contained GDBP-38.

Thank you for this remark. Figure 7 is now labeled figure 8 after adding the supplemental figures per your advice.

We ran a lot of samples (control 12 and TBI 30) and 7A is a representative of some of the samples. The reason that not all the patients will have the GBDP 38K due to patient variability.

Line 628, Figure legend; “Statistical analysis is shown in Figure 9”.  Where is Figure 9???

We removed figure 9 before submission

We apologize for the confusion

Discussion

First 4 paragraphs of Discussion are really just a re-hash of the results.  Consider shorting considerably.

Thank you for this suggestion. We have summarized the paragraphs per your advice.

Line 682 to 684. Be consistent when describing cleavage sites; “cleavage site between D225 and V226” or “cleavage site at ELND78*R79”.  Check throughout manuscript.

We have corrected that throughout the manuscript

Line 715: Is there any evidence indicating GBDPs are released into the blood flowing brain injury, and should the feasibility to routinely measure GBDPs in CSF in the clinic after TBI be discussed?

We thank you for this interesting comment. This hasn't been evaluated yet and would be a very critical point in studying GBDP in CSF as well as its correlation in blood flowing.